# Nicotine Changes Airway Epithelial Phenotype and May Increase the SARS-COV-2 Infection Severity

**DOI:** 10.3390/molecules26010101

**Published:** 2020-12-28

**Authors:** Leonardo Lupacchini, Fabrizio Maggi, Carlo Tomino, Chiara De Dominicis, Cristiana Mollinari, Massimo Fini, Stefano Bonassi, Daniela Merlo, Patrizia Russo

**Affiliations:** 1Molecular and Cellular Neurobiology, IRCSS San Raffaele Pisana, Via di Val Cannuta 247, I-00166 Rome, Italy; leonardo.lupacchini@sanraffaele.it (L.L.); chiaradedominicisjob@gmail.com (C.D.D.); 2Department of Medicine and Surgery, University of Insubria, viale Luigi Borri 57, I-21100 Varese, Italy; fabrizio.maggi63@gmail.com; 3Scientific Direction, IRCSS San Raffaele Pisana, Via di Val Cannuta 247, I-00166 Rome, Italy; massimo.fini@sanraffaele.it; 4Institute of Translational Pharmacology, National Research Council, Via Fosso del Cavaliere 100, 00133 Rome, Italy; cristiana.mollinari@ift.cnr.it; 5Department of Neuroscience, Istituto Superiore di Sanità, Viale Regina Elena 299, I-00161 Rome, Italy; daniela.merlo@iss.it; 6Clinical and Molecular Epidemiology, IRCSS San Raffaele Pisana, Via di Val Cannuta 247, I-00166 Rome, Italy; stefano.bonassi@uniroma5.it; 7Department of Human Sciences and Quality of Life Promotion, San Raffaele University, Via di Val Cannuta 247, I-00166 Rome, Italy

**Keywords:** cell proliferation, EMT, mitochondrial dysfunction, nAChR, nicotine, SARS-CoV-2

## Abstract

(1) Background: Nicotine is implicated in the SARS-COV-2 infection through activation of the α7-nAChR and over-expression of ACE2. Our objective was to clarify the role of nicotine in SARS-CoV-2 infection exploring its molecular and cellular activity. (2) Methods: HBEpC or si-mRNA-α7-HBEpC were treated for 1 h, 48 h or continuously with 10^−7^ M nicotine, a concentration mimicking human exposure to a cigarette. Cell viability and proliferation were evaluated by trypan blue dye exclusion and cell counting, migration by cell migration assay, senescence by SA-β-Gal activity, and anchorage-independent growth by cloning in soft agar. Expression of Ki67, p53/phospho-p53, VEGF, EGFR/pEGFR, phospho-p38, intracellular Ca^2+^, ATP and EMT were evaluated by ELISA and/or Western blotting. (3) Results: nicotine induced through α7-nAChR (i) increase in cell viability, (ii) cell proliferation, (iii) Ki67 over-expression, (iv) phospho-p38 up-regulation, (v) EGFR/pEGFR over-expression, (vi) increase in basal Ca^2+^ concentration, (vii) reduction of ATP production, (viii) decreased level of p53/phospho-p53, (ix) delayed senescence, (x) VEGF increase, (xi) EMT and consequent (xii) enhanced migration, and (xiii) ability to grow independently of the substrate. (4) Conclusions: Based on our results and on evidence showing that nicotine potentiates viral infection, it is likely that nicotine is involved in SARS-CoV-2 infection and severity.

## 1. Introduction

Nicotine (Table 1) [1,2,3,4,5,6,7,8,9] is the addictive compound of tobacco and exerts its effect after binding to nAChR [10,11,12]. ACh, mAChR, VAChT, AChE, BuChE, and ChAT are all components of the CS. ACh is the natural ligand of both nAChR and mAChR.

nAChR pertain to the super-family of Cys-loop ion channel receptors consisting of nine α (from α2 to α10) and three β subunits (from β2 to β4) activated by the physiological ligand ACh or by the non-physiologic ligand nicotine. At least two α subunits are needed in each nAChR pentamer to bind the ligand, since the binding site is localized inside a groove between the extracellular domains formed by an α subunit and an adjacent subunit. Nicotine binds with high affinity to the α_4_β_2_-nAChR with a K_i_ = 1 nM inducing a desensitized state that in turn reduces functional activity of the α_4_β_2_-nAChR [13], and with low affinity binding to the α7 receptor with K_i_ = 1600 nM [14]. This observation is important because means that α_4_β_2_-nAChR are desensitized by nicotine and thus are not able to function. A recent work, using a comprehensive “methyl scan” approach, shows that the orthosteric (i.e., site where the endogenous ligand binds and produces its effects) binding sites for ACh and nicotine in the α_4_β_2_-nAChR and the α7-nAChR interact differently with the pyrrolidinium ring of nicotine, and suggests possible reasons for the higher affinity of nicotine for α_4_β_2_-nAChR [15]. The α_4_β_2_-nAChR are prevalently localized on the brain and mediate many behaviors related to nicotine addiction and are the primary targets for currently approved smoking cessation agents [12,16]. Lung epithelial cells (unaffected and cancer cells) express different nAChR [17]. It has been reported that, although there is a significant reduction in α4-nAChR expression in lung cancer [18], there is so far no further evaluation of its role in normal bronchial epithelium. It has been hypothesized that α7 and α_4_β_2_ nAChR activation may be associated with transactivation via cAMP stimulation of EGFR signaling pathways inducing cell growth and proliferation, thus may contribute to carcinogenesis [19]. The role of the α7 receptor is the best characterized on the lung [12].

Heteromeric nAChR is constituted by α and β sub-units assembled around a central pore (the ionic channel) with different stoichiometry, α7 and α9 are homopentamers. Different reviews have described extensively the nature, structure and function of nAChR [10,11,20]. The presence of functional CS in non-neuronal cells are actually well established [21,22]. Thus, it is now accepted that ACh acts as a main signaling molecule safeguarding the homeostasis of the entire living organisms from protists to mammals with bacteria and *archaea* considered as the starting points of the universal phylogenetic tree [23,24,25,26,27]. The CS is involved in non-neuronal cells, considered non excitable cells (i.e., cells that do not generate action potentials), in different functions such as proliferation, neo-angiogenesis, apoptosis, inflammation and immunity [12,28]. More than 40 types of cells are present on the lung and nAChR, with a predominance of α7-nAChR, are expressed in the majority of these cells [17,29,30,31,32,33,34,35,36]. Recently, nicotine has been implicated in the SARS-CoV-2 infection, since, nicotine, through activation of the α7-nAChR subtype, induces over-expression of ACE2 [35,36,37]. ACE2 is the only confirmed SARS-CoV-2 entry receptor [38,39]. SARS-CoV-2 and ACE2 binding allow the virus to enter into a cell through the spike (S) proteins that work in concert with the host cell TMPRSS2. TMPRSS2 cleaves the spike protein of SARS-CoV-2 facilitating membrane fusion. ACE2 relies on the RAS molecular system, ACE2 is a crucial counter-regulatory enzyme to ACE by the breakdown of angiotensin II that is involved in blood pressure regulation and electrolyte homeostasis [40]. Moreover, ACE2 is involved in the bradykinin metabolism in the lungs (i.e., vasodilation and elevation of vascular permeability) [41]. Recent work shows that there is a variable infection gradient in the respiratory tract. Indeed, through high-sensitivity RNA in situ mapping technology, it has been shown that the highest ACE2 expression is detecting in the nose with a decreased expression all through the lower respiratory tract, this observation is paralleled by an outstanding gradient of SARS-CoV-2 infection in proximal (high) versus distal (low) pulmonary epithelial cultures [42]. Current smoking is associated with increased expression of gene for ACE2, and TMPRSS2 [43,44,45,46]. It has been shown that airway epithelial cell expression of CHRNA7, encoding α7-nAChR, is significantly correlated with the expression of ACE2 with a trend towards higher expression in current smokers as compared with former and never smokers [36]. All together these data support the hypothesis that whenever ACE2 expression increases SARS-CoV-2 acquires more gates of entry; thus, exposure to nicotine via smoking or vaping may enhance susceptibility to SARS-CoV-2 infection. Indeed, we have recently shown that nicotine exposure induces rapid and long-lasting increase in gene expression and protein level of ACE2 in low ACE2-expressing human pulmonary adenocarcinoma A549 epithelial cell line, which in turn translates into increased competence for SARS-CoV-2 replication and cytopathic effect [47]. Different epidemiological investigations on relationship between smoking (or nicotine) and SARS-CoV-2 have been published or are available as pre-print, and we have recently extensively reviewed these data [48]. Evidence from these studies is somehow contrasting, and the role of tobacco smoking or vaping is still to be assessed [49,50,51,52,53,54,55,56,57,58,59,60]. It is possible to suppose the presence of underreporting or misreporting data in recording smoking history at admission, especially to ICU, during the COVID-19 outbreak, when the hospitals are overwhelmed. A recent paper published by Cattaruzza et al. [61] considers that the low prevalence among hospitalized patients are partially due to many smokers misclassified as nonsmokers and concludes that tobacco smoking in a dose-dependent model up-regulates ACE2, and this mechanism may explain the high risk of severe COVID-19 in smokers. The annual meeting of ERS (30th Annual Meeting of the European Respiratory Society; September 7–9, 2020 Wien-Österreich) in the section “Epidemiology-Expert view” plans the special section for Monday 7th afternoon “Smoking, nicotine and COVID-19 myths and facts. What is the evidence?” [47]. To explain the role of tobacco smoking on COVID-19 severity and progression, a longitudinal observational study titled COvid19 and SMOking in Italy (COSMO-IT) was designed [62]. The main objective of the COSMO-IT study is to measure and define the role of tobacco smoking and smoking cessation on the severity and progression of COVID-19 in hospitalized patients. Interestingly, a recent population- study, that surveyed 4351 adolescents and young adults aged 13–24 years, highlighted that the effects of vaping may collide with the risk of COVID-19, thus users are at a five-seven-times increased risk of a COVID-19 diagnosis, compared with non-users (data adjusted for major confounders, such as age, sex, and obesity [63]. It has been recently hypothesized that sub-chronic e-cig exposure induces inflammatory response and dysregulated repair/extracellular matrix (ECM) remodeling, which occur through the α7 nicotinic acetylcholine receptor (nAChR α7) [64].

On the other hand, Changeux et al. [65] advanced the hypothesis that SARS-CoV-2 may enter the body through neurons of the olfactory system and/or through the lung, and no through ACE2. Changeux et al. [65] proposed nAChR as possible target of SARS-CoV-2 and in turn a protective role against SARS-CoV-2 for nicotine. This hypothesis, currently, is not supported by any scientific proof.

In this important scenario for human health, we have explored the cellular and molecular effects induced by nicotine in human bronchial epithelial cells to better understand the nicotine’s role in SARS-CoV-2 entry, disease development, and severity. We explored extensively the biological effects induced by nicotine that are important in SARS-CoV-2 infection, effects not explored in our previous paper [35].

## 2. Results

HBEpC expresses functional α7-nAChR that increases both after short time (1 h) and continuous (1 treatment every 48 h for a total of 16 passages) exposure to nicotine at 1 × 10^−7^ M [35]. HBEpC expresses also ACE2 that, after nicotine treatment, increases concurrently with the increment in α7-nAChR proteins [35]. On the contrary, si-mRNA-α7-HBEpC that does not express α7-nAChR after nicotine treatment does not show any increase in ACE2 proteins [35]. The effect of nicotine 1 × 10^−7^ M on cell viability was evaluated both in the HBEpC and si-mRNA-α7-HBEpC cells treated for 1 h. Cells were counted 48 h after drug removal. We found that nicotine enhances cell viability only in wild-type cells (Figure 1A).

Cell proliferation was evaluated exposing HBEpC and si-mRNA-α7-HBEpC to nicotine over time (from zero to 96 h): Nicotine was added every 48 h. Nicotine significantly increased cell proliferation, however, when α7-nAChR was silenced, cells grow more slowly and nicotine failed to induce cell proliferation (Figure 1B). The doubling time of HBEpC, under our experimental conditions is 44 ± 2 h, when cells were exposed continuously to 1 × 10^−7^ M nicotine, the doubling time was 25 ± 1 h with a reduction of 43.18%. The doubling time of si-mRNA-α7-HBEpC was higher than 96 h and remained the same also after continuous treatment with nicotine (Figure 1B).

Ki67 expression, a marker of proliferation, was evaluated by ELISA in both wild type and silenced cells after treatment with nicotine for 48 h. HeLa cells, an immortal cancer cell line with a doubling time of ~26.67 h, were used as positive control (Figure 2A). The amount of Ki67 in HBEpC cells was 878.24 ± 35 pg/mL in the absence of nicotine, increasing by 117.13% in presence of nicotine (1,907.05 ± 28 pg/mL) with respect to untreated cells. Thus, when nicotine was present the level of Ki67 was very similar to that detected in HeLa cells 1,907.05 ± 28 pg/mL versus 2,149.10 ± 73 pg/mL). si-mRNA-α7-HBEpC showed low level of Ki67 that did not increase after nicotine exposure (212.65 ± 13.3 and 152.13 ± 31.8, respectively). These findings were confirmed also in Western blotting experiments (Figure 2C,D).

HBEpC has a finite lifetime in cell culture before dying, thus, starting from the 16th passages HBEpC grew more slowly than their counterpart growing in the continuous presence of nicotine. At the 24th HBEpC stopped to grow and divide while nicotine-treated HBEpC retained the ability to divide and grow (data not shown). The extent of senescence is correlated with the increased activity of the SA-β-Gal. Accordingly, to cell growth data, SA-β-Gal levels started to increase for the 8th passage reaching the maximum level the 24th passage, in a time-dependent manner. In the presence of nicotine, SA-β-Gal level started to increase at the 16th passage, not as much as in untreated cells (−37.7% *p* < 0.001), and at the 24th, its amount was 28.56% lower than in untreated cells (*p* < 0.001) (Figure 3).

Moreover, we found that following 48 h nicotine treatment, the basal levels of Ca^2+^ in HBEpC significantly increased (*p* = 0.0004) from 3.9 ± 0.3 mM to 7.9 ± 0.2 mM, but no in the silenced cells (2.6 ± 0.3 versus 2.9 ± 0.4 mM) (Figure 4).

Under the same experimental condition, nicotine decreased the amount of ATP in HBEpC from 6.04 ± 0.14 to 0.63 ± 0.12 nM (*p* =0.0003) (Figure 5).

We evaluated also the expression of human EGFR and pEGFR in 48 h nicotine treated HBEpC, which reached the level of 0.56 ± 0.11 and 1.33 ± 0.31 ng/mL, increasing by 138% with respect to untreated cell (Figure 6A,B) and 0.58 ± 0.08 IU/mL and 1.14 ± 0.09 IU/mL with an increase of 97.2%, respectively (Figure 6C,D). si-mRNA-α7-HBEpC showed a low level of both EGFR and pEGFR that did not increase after nicotine exposure (Figure 6).

Nicotine decreased the amount of p53 and phoshop53 proteins. In the ELISA test, p53 was reduced by 24.42% after 1 h and by 53.49% after 48 h, whereas phospho-p53 was reduced by 43.24% both after 1 and 48 h. The Western blotting confirms the reduction of p53 in a time-dependent manner (400 μM H_2_O_2_ positive control) (Figure 7).

Nicotine increased the amount of phosho-p38 proteins after 1 h of treatment (Figure 8A,B). The amount of phosho-p38 in untreated cells was 10.70 ± 1.11 U/mL and in treated cells was 24.2 ± 2.43 U/mL with an increase of 126%. No variation of p38 was observed (Figure 8A,C).

Nicotine treatment for 48 h induces EMT characterized by inhibition of E-Cadherin expression and up-regulation of mesenchymal markers such as FN and Vimentin (Figure 9).

HBEpC can migrate, although with a lower extent than HeLa cells (positive control known to be able to migrate Figure 10A). Nicotine exposure for 48 h in complete Bronchial/Tracheal Epithelial Cell Growth Medium significantly (*p* < 0.0001) increased the migration ability (+74.6%). The effect was strictly dependent on α7-nAChR, since si-mRNA-α7-HBEpC abolished their ability to migrate (Figure 10B).

HBEpC produce VEGF. nicotine exposure for 48 h significantly (*p* < 0.0001) increases the release of VEGF namely from 696.28 pg/mL to 1,409.9 pg/mL with 102%. The effect is strictly dependent on α7-nAChR, since in si-mRNA-α7-HBEpC no increase is observed (Figure 11).

Anchorage-dependent growth (ADG) is mandatory for non-transformed epithelial cell survival. Thus, HBEpC is not able to grow when seeded onto soft agar (Figure 12A). HBEpC cells at the 16th passage (one passage every 48 h) under the continuous presence of nicotine showed the ability to grow independently of the substrate (Figure 12). However, treated cells in comparison with HeLa cells (positive control) grew ~62% less. This effect was not observed in si-mRNA-α7-HBEpC that grow similarly to NIH3T3 (negative control or untreated HBEpC).

## 3. Discussion

In this work, we report data to support our hypothesis that nicotine may be involved in the processes facilitating SARS-CoV-2 infection [35]. We have previously shown in HBEpC that nicotine enhances the expression levels of α7-nAChR and ACE2 through MAPK/ERK activation [35]. Moreover, nicotine increases SARS-CoV-2 replication, transcription of SARS-CoV-2 viral proteins, and SARS-CoV-2 cytopathic effect [47]. Thus, it is reasonable to assume that the entrance and replication of SARS-CoV-2 may be caused by the mechanisms related to nAChR mediated-signaling, as previously hypothesized [35,37]. In this work, our data show that nicotine, under conditions mimicking the human exposure in smokers (i.e., concentration and exposure time), through induction of α7-nAChR induces: (i) increase in cell viability; (ii) cell proliferation; (iii) Ki67 over-expression; (iv)phospho-p38 up-regulation; (v) EGFR and pEGFR over-expression; (vi) more basal Ca^2+^ concentration; (vii) reduction of ATP production; (viii) decreased level of p53 and phosphor-p53; *(ix)* delayed senescence; (x) VEGF increase; (xi) EMT and consequent; (xii) enhanced migration; and (xiii) ability to grow independently of the substrate. Some of these characteristics are typical of cancer cells [66]. All these effects are strictly dependent by α7-nAChR. Thus, when α7-(si-mRNA-α7-HBEpC) or α4-nAChR were silenced (si-mRNA-α4-HBEpC) no ACE2 upregulation was observed (see Appendix A for si-mRNA-α4-HBEpC).

These results may help to understand the causal link between exposure to nicotine and the progression and severity of SARS-CoV-2 infection. Table 2 shows the effects induced by nicotine, the effects induced by SARS-CoV-2, by SARS-CoV, and MERS-CoV as well as by non-tumorigenic virus infection

HBEpC is derived from the surface epithelium of unaffected (normal) human bronchi consequently they have a finite lifetime in cell culture before dying, similarly to all cultures of adult human alveolar epithelial type II (AT2) cells which are characteristically short-lived and feeder-dependent [110]. As cells approach the limit, commonly known as the “Hayflick limit” [111], cells start to show different signs of aging and then die. In the presence of nicotine, HBEpC can grow for a longer time (more than 24 passages equal to 48 days) also independently of the substrate (soft agar). In brief, nicotine prevents or delays cell death by making more functional cells available for a possible viral infection. It has been shown that SARS-CoV-2 rapidly replicates in actively transcriptional cells causing major readjustments in cellular functions, including splicing, proteostasis, and nucleotide biosynthesis [78]. Indeed, Coronavirus replication depends on the availability of cellular nucleotide pools [79], and consequently of a proliferating and viable cell. SARS-CoV-2 protein interaction map [112] highlights interactions between SARS-CoV-2 proteins and human proteins that are involved in several complexes and biological processes including DNA replication (i.e., NSP1). The SARS-CoV-2 NSP1 is known to suppress host gene expression by ribosome association [80].

Several data imply p53 wild-type in the viral life cycle of non-tumor-promoting viruses, indeed some viruses need active p53 for their efficient replication, whereas others need reduction/inhibition of its activity. The IAV infection is strictly p53-dependent: in p53-deficient mice, IAV induces higher mortality, and higher viral load in the lungs than in the p53 counterparts. The knockdown of p53 expression by RNA interference enhances IAV replication. All data suggest that the absence of p53 may delay the innate response, causing severe IAV-induced morbidity as observed in the p53KO mice [89]. In SARS-CoV disease, p53 works as an anti-viral factor inhibiting viral replication thus in cells lacking p53 the rate of virus replication is higher than in cells expressing p53 [88].

A computational analysis of microRNA-mediated interactions in SARS-CoV-2 infection revealed that among different affected pathways that of the of EGFR (17 genes) is influenced by SARS-CoV-2 [84]. It has been shown that in human bronchial epithelial cells treated with SARS-CoV, for 12, 24, and 48 h, the expression of EGFR gene is increased 12 h after the infection and then decreases after 24–48 h [85]. SARS-CoV-2 Spike and SARS-CoV Spike bind with similar affinities to human ACE2 [113].

More Ca^2+^ elevation and reduced ATP production suggest a mitochondrial dysfunction induced by nicotine. As well known, mitochondria express α7-nAChR; nicotine in isolated mitochondria decreases the consumption of O_2_ and the mitochondrial membrane potential whereas in intact cells the consumption of O_2_, and the Ca^2+^ levels increase. The reduction of the mitochondrial membrane potential is linked to ATP reduction [92]. High levels of cytosolic Ca^2+^ concentrations and influx of Ca^2+^ into mitochondria sustain viral replication of different respiratory viruses such as RSV, MV RV [98], and SARS-CoV [97]. A recent study shows that over a two-fold increase in intracellular Ca^2+^enhances the enter ability of the MERS-CoV. In the absence of Ca^2+^ fusion is attenuated, but not completely abrogated [96]. On the other hand, depletion of intracellular Ca^2+^ completely abrogates SARS-CoV host cell entry [97]. It has been shown that during viral pathogenesis deregulation of Ca^2+^ homeostasis alters both mitochondrial dynamics [114,115] and mitochondrial metabolic pathways to sustain cellular energy homeostasis that in turn ensure efficient virus replication and prevent mitochondrial antiviral response [116].

It has been found, recently, that human lung adenocarcinoma cell line A549, transduced with human ACE2 (hACE2), and then infected with SARS-CoV-2 show perturbations in different pathways including down regulation of genes in the mitochondrial and electron transport chain processes. Similar alterations are observed in infected human nasopharyngeal samples, used as control [93]. When we exposed A549 cells treated with nicotine to SARS-CoV-2 we found a very high virus replication without transduction of ACE2 in these cells, thus nicotine was able to increases both mRNA and protein for ACE2 allowing the entry of the virus in a big amount [47].

EMT is a process where epithelial cells, including those lining the lung mucosa, lose their polarity and adhesion. Thus, EMT provides migratory and invasive properties to cells. EMT develops in different pathologies including viral infections. Vimentin is one player involved into EMT. Vimentin, expressed mainly in cells of mesenchymal origin, acts together with other cytoskeletal components modulating cell migration, adhesion and division [117,118,119,120]. Additionally, Vimentin plays important roles during viral infection and replication cycles [121]. It has been reported an important role for Vimentin in SARS-CoV virus entry through interaction with its S protein [105] in the SARS-CoV-permissive cell line Vero E6 (African green monkey kidney epithelial cells). In these cells the expression of Vimentin is up-regulated after virus interaction and enhances its delivery to ACE2. Although SARS-CoV-2 is ~82% identical to human SARS-CoV and ~50% to MERS-CoV [122], SARS-CoV-2 shows unique features, thus the above results remain to be confirmed in human cells. However, a recently interaction map that includes all proteins of SARS-CoV-2 identified a potential interaction between several viral proteins, including S protein and Vimentin [123]. In light of the above considerations, it has been speculated that drugs able to decrease the expression of Vimentin may be used for the treatment of patients with COVID-19 [120]. On the other hand, drugs that increase Vimentin expression such as nicotine may be considered extremely dangerous by increasing SARS-CoV-2 infection ability. Bioinformatics analysis shows that VEGF and FN interact with ACE2, the mRNA and protein of these molecules are more expressed in lung epithelial cells and also after SARS-CoV-2 infection [103]. Not all SARS-CoV-2 infections evolve into severe COVID-19. Looking at the different steps of SARS-CoV-2 infection ahead the lung injury (first hit), the severity of the disease is determined by the host response [124]. Indeed, the lung infection, in some patients, leads to increased expression of VEGF [125]. VEGF is a key stimulator of angiogenesis. Angiogenesis, or better neo-angiogenesis, is a physiological and well-regulated process where new blood vessels are formed from pre-existing vessels [126]. Deregulated angiogenesis is observed in pathological situations (i.e., cancer). Viruses, such as CMV or HCV, may regulate angiogenesis directly or indirectly, activating vessels through endothelial cell tropism and/or producing chemokines and/or growth factors (i.e., VEGF) creating a pro-angiogenic microenvironment [108]. Morphologic and molecular features of lungs, obtained at autopsy from patients who died from SARS-CoV-2 infection show the presence of IA as well as of conventional SA [107]. IA, also called splitting angiogenesis or non-sprouting angiogenesis, is currently considered as an important alternative and complementary form of SA characterized by the presence of the so-called intraluminal tissue pillars formed by an invagination of the capillary walls into the vascular lumen [127]. Nicotine activates not only MAPK, EGFR, and VEGF pathways, but also Rb-E2F and JAK-STAT signaling pathways and its target genes [83,128,129]. Specifically, STAT3 up-regulation is α7-nAChR dependent. JAK-STAT is activated via EGF and VEGF [83]. Infection by SARS-CoV-2 delivers into cells NSP1 and ORF6, which efficiently inhibit STAT1 function that in turn increases STAT3 activity. The aberrant transcription in the direction of STAT3 may lead to the catastrophic cascades specific for COVID-19 patho-physiologies such as rapid coagulopathy/thrombosis, proinflammatory conditions, profibrotic status, and T cell lymphopenia in infected patients (reviewed in [130]).

All the observations reported by our experiments, in agreement with most literature, (Table 2) imply a strong involvement of nicotine in SARS-CoV-2 infection. In this context it is important to remind that secreted human SLURP-1 works as an auto/paracrine regulator of physiological processes indeed inhibits selectively ACh-evoked current through the α7-nAChR and suppresses the nicotine-induced up-regulation of the α7-nAChR expression and in turn the down-stream pathways. Interestingly, inhibitory analysis revealed that besides α7-nAChR, the antiproliferative effect of SLURP-1 in transfected A549 cells is mediated by EGFR and β-arrestin [128,129]. SLURP-1 shares structural homology with three-finger snakes α-neurotoxins [128,129]. Thus, SLURP-1 may protect against SARS-CoV-2.

Nevertheless, some authors believe that nicotine may protect against SARS-CoV-2 [47,48,71,131]. The fundamental hypothesis behind this statement is that nicotine reduces inflammation through its interaction with the nAChR and that nicotine itself interacts with SARS-CoV-2 inhibiting virus attachment to ACE2 [65,67,132]. However, the overwhelming evidences showing that nicotine potentiates cardiopulmonary diseases and may enhance viral infection make unlikely a possible therapeutic benefit in COVID-19 due to nicotine, and rather push to strengthen policies against tobacco smoking also in view of a possible SARS-CoV-2 infection.

## 4. Materials and Methods

### 4.1. Cells and Treatment

Human Bronchial Epithelial Cells (HBEpC) were obtained from Cell Applications Inc. (www.cellapplications.com/product no. 502K-05a) and cultured in complete Bronchial/Tracheal Epithelial Cell Growth Medium (www.cellapplications.com/product) as described previously [35]. si-mRNA-α7-HBEpC were obtained as described previously [35]. A total of 7.5 × 10^4^ cells/cm^2^ semi-confluent cells were treated (a) for 1 h or 48 h with zero or 1.0 × 10^−7^ M nicotine (Sigma-Aldrich, Milan, Italy) dissolved in saline in complete medium; (b) treated continuously with nicotine for additional passages, 1 passage every 48 h for a total of 16 or 24 passages.

### 4.2. Western Blot Detection

Cells were lysed using complete lysis buffer (Roche, Monza, Italy, www.roche.it) plus protease inhibitor cocktail (PIC, Complete-M, Roche). The protein concentration was determined using the BCA protein assay (Roche). After mixing with Laemmli’s buffer, samples were subjected to SDS-PAGE and Western blotting. For immunodetection, the following antibodies were used: anti-p53 (DO-1): sc-126, anti-Vimentin(V9): sc-6260, anti-Fibronectin (EP5): sc-8422, anti-E-cadherin (G-10): sc-8426, anti-Ki67 (Ki-67): sc-23900, ACE2(E-11): sc-390851 and α4-nAChR (A-6): sc-74519 (Santa Cruz Biotechnology, Inc, Dallas, TX, USA), anti-β-actin catalog number: A3853 (Sigma-Aldrich Italia). Horseradish peroxidase-labeled anti-mouse or anti-rabbit secondary antibodies catalog: 711-035-152 (Jackson, Cambridge, UK) and an enhanced chemiluminescence kit (Western blot detection reagent, GE Healthcare UK Limited, Amersham, UK) were used for the detection of recognized proteins. Primary antibody was diluted 1:1000 and secondary 1:10,000. Densitometric analysis for quantification of the relative level of protein expression was performed using Amersham Image Quant800 (EG Healthcare) with software ImageQuant TL 7.0^®^.

### 4.3. Evaluation of p53/phosphor-p53, p38/phospo-p38, Calcium, ATP, EGFR/p-EGFR, and VEGF

PathScan^®^ apoptosis multi-target sandwich ELISA kit (Cell signaling technology) was used to detect p53 and phospho-p53 (pP53). Calcium detection assay kit (colorimetric) ab102505 was purchased from Abcam (Biotech, Life sciences, Cambridge, UK). ATP colorimetric assay kit (k354-100) was purchased from BioVision (Milpitas, CA, USA). Human EGFR (pY1068) ELISA kit (KHR9081) and Human EGFR (Full length) ELISA kit (KO-IR9061) were purchased from Invitrogen (Thermo-Fisher, Waltham, MA, USA). Human VEGF QuantiGlo ELISA Kit (QVE00B) was purchased from R&D Systems, Inc. (Minneapolis, MN, USA). STAR phospho-p38α (Thr180/Tyr182) ELISA Kit: 17-488 was purchased from Merck-Sigma Aldrich (Milan, Italy); p38 MAPK (Total) Human ELISA Kit (KHO0061) were purchased from Invitrogen (Thermo-Fisher, Waltham, MA, USA). All experiments were performed according to the manufacture’s protocol.

### 4.4. Cell Migration

CytoSelect^®^ 24-Well Cell Migration Assay (8 μm, Fluorometric format) was obtained from Cell Biolabs (catalog number CBA-101-C (San Diego, CA, USA)) and experiments were performed according to the manufacture’s protocol.

### 4.5. Cellular Senescence

The 96-Well Cellular Senescence Assay Kit (SA-β-Gal Activity, Fluorometric Format) was obtained from Cell Biolabs (catalog number CBA-231 (San Diego, CA, USA)) and experiments were performed according to the manufacture’s protocol.

### 4.6. Cell Transformation

CytoSelect^TM^ 96-well cell transformation Assay (CBA-140) was purchased from Cell Biolabs Inc. (San Diego, CA, USA) and experiments were performed according to the manufacture’s protocol.

### 4.7. Statistical Analysis

Data were managed and analyzed using GraphPad Prism 8.1^®^ (GraphPad Software Inc., La Jolla, CA, USA); two tails paired *t*-test or one-way ANOVA with multiple-comparison and post hoc test with Bonferroni correction were used to evaluate statistical significance. Data were described as mean ± SD (standard deviation). A *p*-value of <0.05 was considered as statistically significant. Each experiment was performed at least two times in separate experiments done at least in triplicate.

## 5. Conclusions

Based on our results and on evidence showing that nicotine potentiates viral infection it is likely that nicotine is involved in SARS-CoV-2 infection and severity

## Figures and Tables

**Figure 1 molecules-26-00101-f001:**
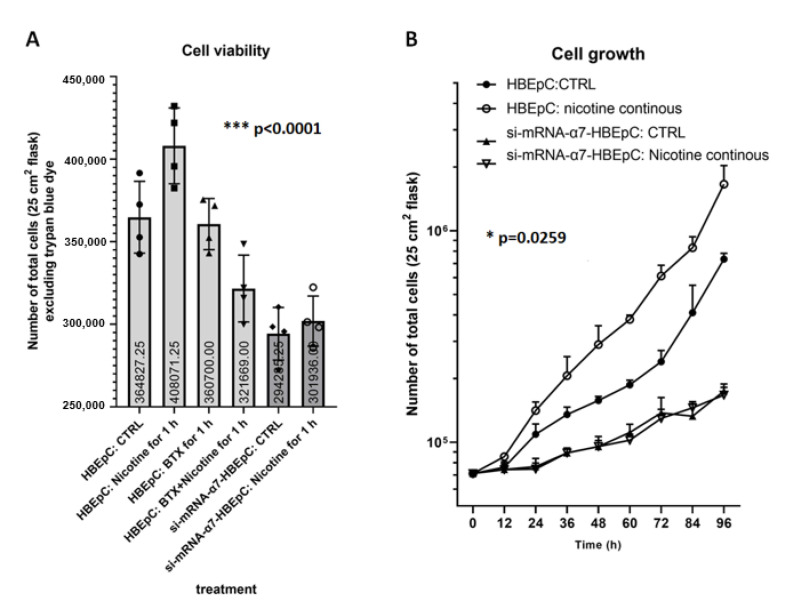
Cell viability (**A**) and cell proliferation (**B**) induced by nicotine in HBEpC and/or si-mRNA-α7-HBEpC viability. (**A**): For cell viability, 7500 cells/cm^2^ are plated in T25 flask (total cell number 187,000) and treated with nicotine 1 × 10^−7^ M, after 1 h cells are washed three times in PBS Ca^2+^ and Mg^2+^ free and then incubated in drug-free medium for additionally 48 h. Then, cells (detached or floating) are counted after staining with trypan blue dye. (**B**): For cell proliferation 7500 cells/cm^2^ are plated in T25 flask and treated with nicotine every 48 h. Cells are detached and viable cells are counted every 12 h. Experiments are performed at least two times in triplicate. Statistical significance is analyzed with one-way ANOVA with multiple-comparison and post hoc test with Bonferroni correction.

**Figure 2 molecules-26-00101-f002:**
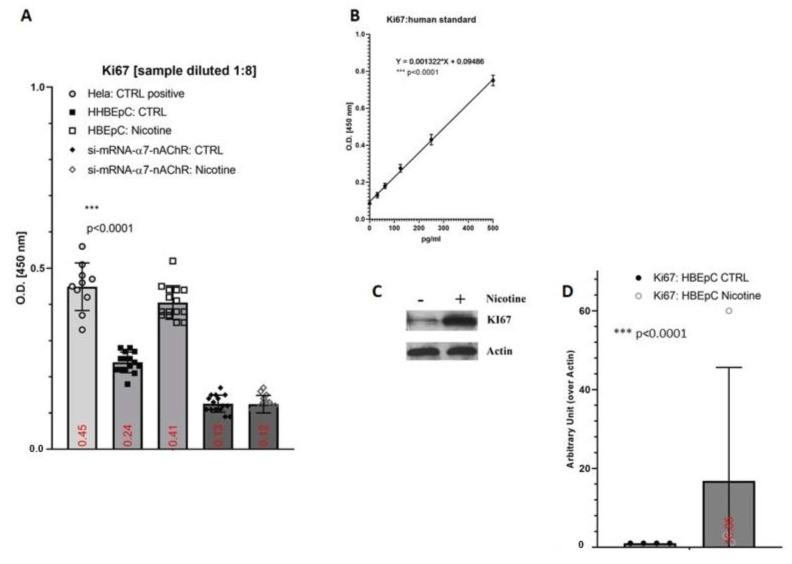
Expression of Ki67 induced by nicotine in HBEpC and/or si-mRNA-α7-HBEpC. (**A**): ELISA experiments; (**B**): regression equation linearity, performed with Prism; (**C**): Western blotting, (**D**): densitometric analysis. Statistical significance is analyzed with one-way ANOVA with multiple-comparison and post hoc test with Bonferroni correction. Experiments are performed at least two times in triplicate. In the Appendix A, raw data of Western blotting are reported.

**Figure 3 molecules-26-00101-f003:**
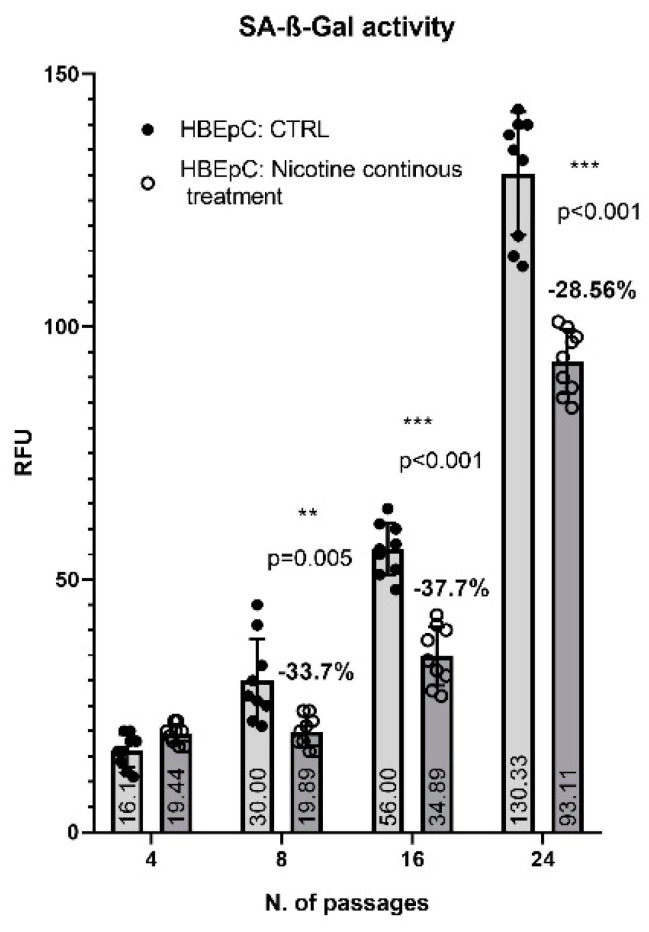
SA-β-Gal in HBEpC exposed continuously to nicotine. Experiments are performed at least two times in triplicate.

**Figure 4 molecules-26-00101-f004:**
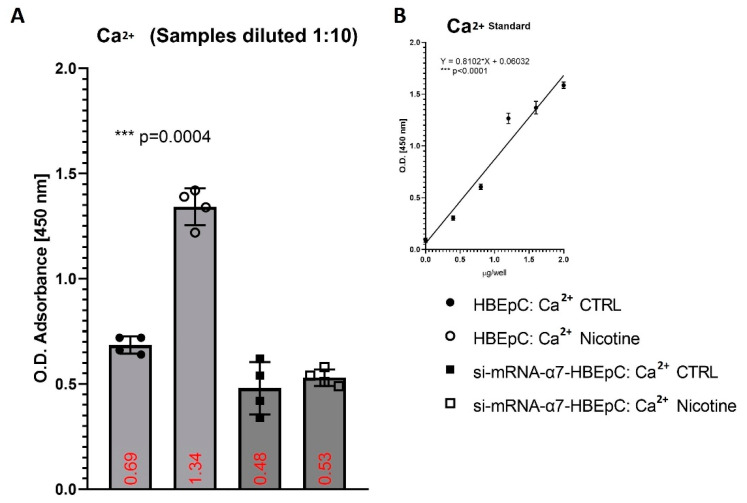
Evaluation of intracellular Ca^2+^ after exposure to nicotine for 48 h in HBEpC and/or si-mRNA-α7-HBEpC. (**A**): ELISA experiments; (**B**): regression equation linearity, performed with Prism. Statistical significance is analyzed with one-way ANOVA with multiple-comparison and post hoc test with Bonferroni correction. Experiments are performed at least two times in triplicate.

**Figure 5 molecules-26-00101-f005:**
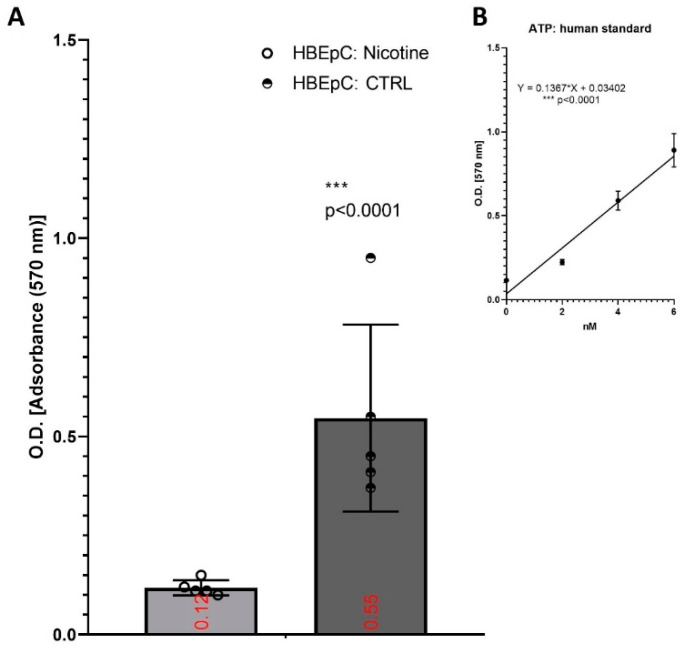
Evaluation of intracellular ATP after exposure to nicotine for 48 h in HBEpC. (**A**): ELISA experiments; (**B**): regression equation linearity, performed with Prism. Experiments are performed at least two times in triplicate.

**Figure 6 molecules-26-00101-f006:**
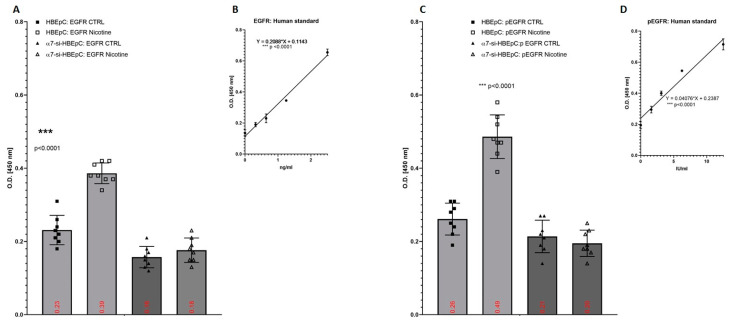
Expression of EGF and p-EGFR after exposure to nicotine for 48 h in HBEpC. (**A**): ELISA experiments for EGFR; (**B**): regression equation linearity for EGFR, performed with Prism; (**C**): ELISA experiments for p-EGFR; (**D**): regression equation linearity for p-EGFR, performed with Prism. Statistical significance is analyzed with one-way ANOVA with multiple-comparison and post hoc test with Bonferroni correction. Experiments are performed at least two times in triplicate.

**Figure 7 molecules-26-00101-f007:**
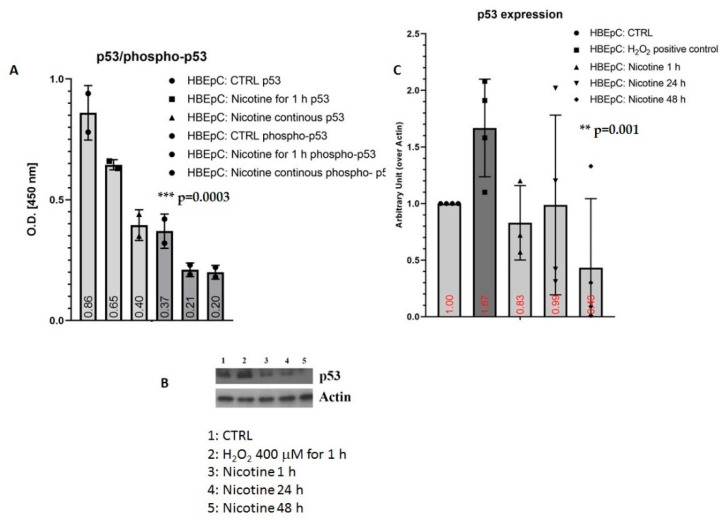
Induction of p53 and phospho-p53 induced by nicotine in HBEpC. (**A**): ELISA assay. (**B**): Western blotting experiments, (**C**): densometric analysis. Experiments are performed at least two times in triplicate. Statistical significance is analyzed with one-way ANOVA with multiple-comparison and post hoc test with Bonferroni correction. In the Appendix A, raw data of Western blotting are reported.

**Figure 8 molecules-26-00101-f008:**
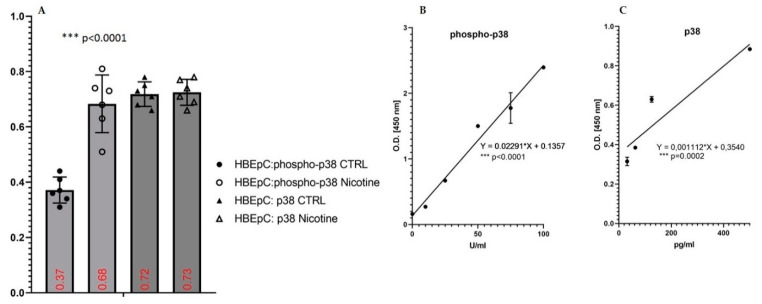
Induction of phospho-p38 and p38 by nicotine in HBEpC. (**A**): ELISA experiments; (**B**): regression equation linearity for phosphor-p38; (**C**): regression equation linearity for phosphor-p38performed with Prism. Statistical significance is analyzed with one-way ANOVA with multiple-comparison and post hoc test with Bonferroni correction. Experiments are performed at least two times in triplicate.

**Figure 9 molecules-26-00101-f009:**
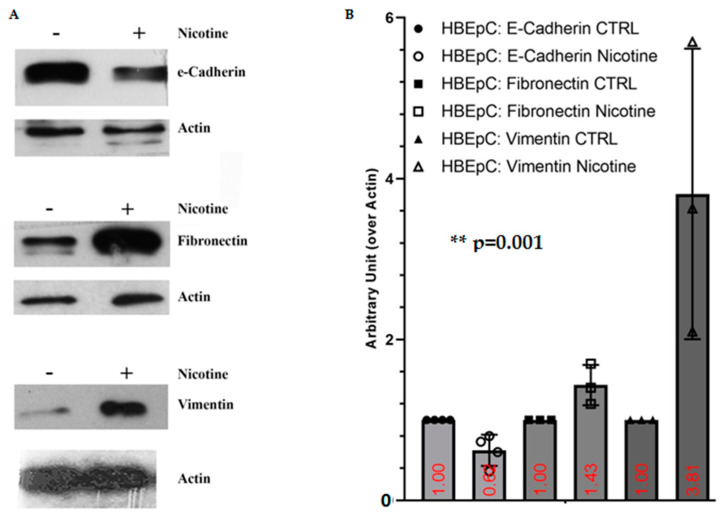
EMT induced by nicotine in HBEpC. (**A**): Western blotting; (**B**): densitometric analysis. Statistical significance is analyzed with one-way ANOVA with multiple-comparison and post hoc test with Bonferroni correction. Experiments are performed at least two times in triplicate. In the Appendix A, raw data of Western blotting are reported.

**Figure 10 molecules-26-00101-f010:**
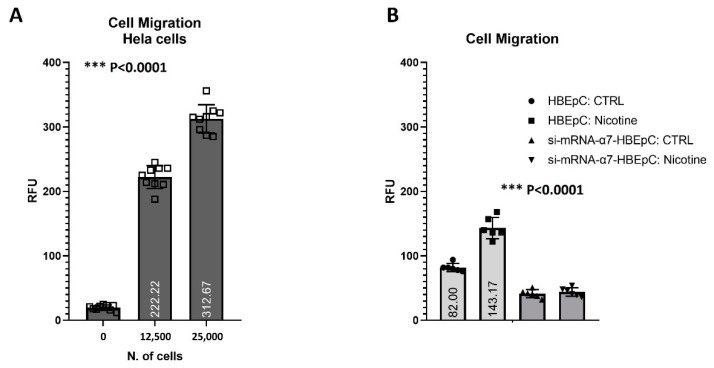
Cell migration to nicotine for 48 h in HBEpC and/or si-mRNA-α7-HBEpC. HeLa cells are positive control. (**A**) HeLa positive control. (**B**): Cell migration h in HBEpC and/or si-mRNA-α7-HBEpC. Statistical significance is analyzed with one-way ANOVA with multiple-comparison and post hoc test with Bonferroni correction. Experiments are performed at least two times in triplicate.

**Figure 11 molecules-26-00101-f011:**
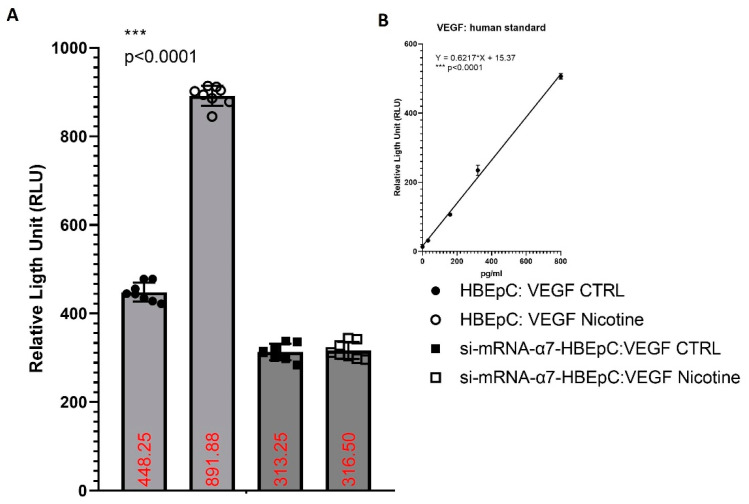
Induction of VEGF by nicotine in HBEpC and/or si-mRNA-α7-HBEpC. (**A**): ELISA experiments; (**B**) regression equation linearity for VEGFR. Statistical significance is analyzed with one-way AOVA with multiple-comparison and post hoc test with Bonferroni correction. Experiments are performed at least two times in triplicate.

**Figure 12 molecules-26-00101-f012:**
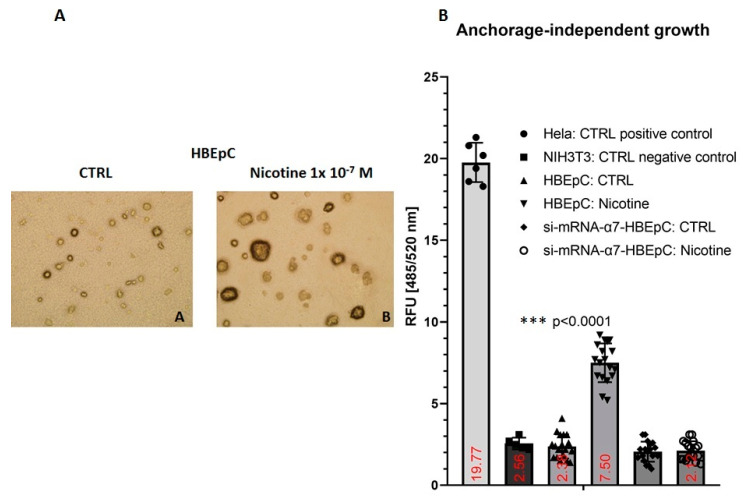
Anchorage-independent growth induced by nicotine in HBEpC. HeLa cells are positive control and NIH3T3 the negative. (**A**): Representative picture of HBEpC cloned on soft agar. (**B**): Cloned cells. Statistical significance is analyzed with one-way ANOVA with multiple-comparison and post hoc test with Bonferroni correction. Experiments are performed at least two times in triplicate.

**Table 1 molecules-26-00101-t001:** Nicotine: Chemical, physical, biological characteristics and historical data.

Common Name	Nicotine
IUPAC Name	3-(1-methyl-2-pyrrolidinyl)pyridine
Chemical Specification	Bicyclic molecule characterized by a pyridine cycle and a pyrrolidine cycle existing in natures only in the S shape (i.e., levogyre) [1]
Chemical Characteristics	Chemical formula: C_10_H_14_N_2_ Molecular Weight: 162.234.
Physical Characteristics	Light yellow liquid that turns dark-brown after exposure to light/air with fishy odor when warm. Boiling point: 274.58 °C at 760 Tor [2]
Synthesized as SM by plants	*Familia: Solanaceae*, *Genus: Nicotiana*, *Species*: *Nicotiana tabacum*
History	1492 Cristoforo Colombo discovery of the plant *Nicotiana*. Nicotine owes is name to Jean Nicot (1530-1604) who introduced the use of tobacco to the French court (Caterina de Medici) in the sixteenth century, thus helping the spread of tobacco into all Europe [3] 1828 the German chemists Wilhelm Heinrich Posselt and Karl Ludwig Reimann were the first to isolate nicotine from tobacco. 1843 Louis Melsens described the chemical formula. 1893 Adolf Pinner and Richard Wolffenstein described the structure. 1904 Amé Pictet and Arnold Rotschy, synthesized nicotine [3]
Production, Use and Natural Resistance	Plants produce SM, metabolites not essential for plant reproduction, as direct defenses against pathogens (i.e., insects) and herbivores. Among SM, nicotine is one of the best-studied drug representing one of the first insecticides utilized to control pests in agriculture [4]. Although nicotine is very toxic, some insects develop resistance, indeed the tobacco hornworm *Manduca sexta* may survive to nicotine concentrations that are considered toxic for non-adapted herbivores [4]. Although the precise mechanisms for *Manduca sexta*’s nicotine resistance are not completely understood, an efficient excretion and metabolism seem to be involved. *Manduca sexta* uses nicotine to protect herself by the attack of its native major nocturnal predator wolf spider *Camptocosa parallela* [5]
Concentration in Tobacco leaves	The percentage of nicotine on dry weight of tobacco is between 0.3 to 8.3% depending on plant variety, as for *Nicotiana gossei* and *Nicotiana velutina*, respectively, and 6.7% in Virginia variety, and cultivation [6]
Lethal Doses	In animals the lethal dose may be from 3 mg/kg in mice to 50 mg/kg in rats. There is no consensus on the human lethal dose of nicotine and 60 mg of nicotine given orally (resulting in a plasma concentration of ~180 ng/mL) is often suggested as the lethal dose in the literature. However, it has been reported that adults may survive to dosages much higher than this (up to 500 mg) [7]
Metabolism	Nicotine is metabolized in the liver, principally to cotinine that in turn is metabolized to *trans*-3′-hydroxycotinine excreted via renal [8]
Nicotine concentrations into cigarette	The amount of nicotine in one tobacco cigarette is approximately 1–2% = 1–2 g/100. Considering the human body weight average equal to 68 kg one cigarette may deliver approximately 10–30 μg/Kg, resulting in a peak plasma level of 10–50 ng/mL. A concentration equal to 50 ng/mL can be converted to molarity dividing by nicotine MW (i.e., 162) (50 ng/mL divided by 162) = 0.309 = 3.1 × 10^−7^ M [9]. This concentration is 3.6 times lesser than the lethal dose. 60 mg dose means a 0.8 mg/kg in humans equivalent approximately to the amount found in five cigarettes.

**Table 2 molecules-26-00101-t002:** Effects induced by nicotine on different pathways in human airway epithelial cells (results obtained in this work and in literature) and comparison with effects caused by SARS-CoV-2, SARS-CoV, MERS-CoV and by non-tumorigenic virus infection on the same pathways.

	Effects induced by Nicotine in Human Airway Epithelial Cells [References]	Effects Induced by SARS-CoV-2 [References]	Effects Induced by SARS-CoV or MERS-CoV [References]	Effects Induced by Non-Tumorigenic Virus Infection [References]
**α7-nAChR Down-Stream Pathways**				
α7-nAChR	Increase in HBEpC and A549 cells [35,47]	Silico experiments show the possibility that SARS-CoV-2 may interact with nAChR [67,68]. Nemantine, α7-nAChR antagonist, decreases ACE2 and reduces oxidative stress and inflammation. Nemantine may reduce SARS-CoV-2 virulence [69].		HIV-gp120 induces and regulates mucus formation on airway epithelial cells through a CXCR4-α7-nAChR-GABAAR dependent pathway. Nicotine enhances production of HIV of in vitro-infected alveolar macrophages from healthy cigarette smokers. Nicotine-treated microglia show increased HIV-1 expression in a concentration dependent manner In neuronal cells the α7-nAChR is upregulated after gp120 exposure (Reviewed in [70]).
ACE2	Increase in HBEpC and A549 cells [35,47] Correlation between α7-nAChR and ACE2 [36] and reviewed in [71]	Bioinformatics modeling and in vitro experiments indicate that SARS-CoV-2 utilizes ACE2 as a receptor to gain entry into human cells [39].	It is the entry receptor for SARS-CoV and HCoV-NL63 [72]. The expression of ACE2 is increased 24 h after SARS-CoV infection and remains at a high level after 48 h [73].	
ERK/MAPK, Phospho-p38	Increase in HBEpC (this work and [35,47] and reviewed in [12])	Activation of MAPK signaling [37] Activation of the p38/MAPK pathway during SARS-CoV-2 infection in ACE2-A549 and NHBE cells demonstrated that transcription factors regulated by the p38/MAPK pathway were among the most highly activated upon infection [74] SARS-CoV-2 activates p38 MAPK activity via suppression of NF-κB signaling pathway [75].	SARS-CoV was shown to express proteins that directly upregulate p38 MAPK in vitro [76].	
**Cell Proliferation and Cell Cytotoxicity**				
Cell proliferation	Increase in HBEpC and A549 cells (this work and [35,47] and reviewed in [12,77])	Rapidly replicates in actively transcriptional cells causing major readjustments in cellular functions, including splicing, proteostasis and nucleotide biosynthesis [78]. Replication depends on the availability of cellular nucleotide pools [79]. Interactions between SARS-CoV-2 proteins and human proteins that are involved in several complexes and biological processes including DNA replication (NSP1) [80]. SARS-CoV-2 infects epithelial ACE2- and TMPRSS2-positive cells in the aero digestive and respiratory tracts that are metabolically-primed for glutamine synthesis and rapid replication [81].		
Cell Cytotoxicity	Decrease in HBEpC and A549 cells (this work and [35,47])			
Doubling time	Decrease in HBEpC and A549 cells (this work and[35,47])	A significant increase in the fraction of cells in S phase and at the G_2_/M transition and a decrease in the fraction of cells in G_0_/G_1_ phase were observed [74].		
Ki67	Increase in HBEpC (this work)			
EGFR/pEGFR	Increase in HBEpC (this work and [82,83])	Pathways of EGFR is influenced by SARS-CoV-2 [84].	In human bronchial epithelial cells exposed to SARS-CoV, for 12, 24, and 48 h the expression of EGFR gene is high 12 h after the infection and then decreases after 24–48 h [85].	
p53/phospho-p53	Decrease in HBEpC and A549 cells (this work and [35,47,86,87])		p53 works as an anti-viral factor inhibiting viral replication thus in cells lacking p53 the rate of virus replication is higher than in cells expressing p53 [88].	IAV infection is strictly p53 dependent: in p53-deficient mice IAV induces higher mortality, and higher viral load in the lungs than in p53 counterparts. Knockdown of p53 expression by RNA interference enhances IAV replication. All data suggest that the absence of p53 may delay the innate response, causing severe IAV-induced morbidity as observed in the p53KO mice (Reviewed in [89])
SA-β-Gal activity	Delay in senescence (this work)			
**Mitochondrial Dysfunction**				
	Induction in HBEpC (this work and [90,91,92])	Down regulation of genes in the mitochondrial and electron transport chain processes. Similar alterations are observed in infected human nasopharyngeal samples, used as control [93].		
ATP	Decrease in HBEpC (this work and [92])			
Ca^2+^	Increase basal Ca^2+^ in HBEpC (this work and [94,95])		Over a two-fold increase in intracellular Ca^2+^ enhances the enter ability of the MERS-CoV, in the absence of Ca^2+^ fusion is attenuated, but not completely abrogated [96]. Depletion of intracellular Ca^2+^ completely abrogates SARS-CoV host cell entry [97].	High levels of cytosolic Ca^2+^ concentrations and influx of Ca^2+^ into mitochondria sustain viral replication of different respiratory viruses such as RSV, MV and RV [97,98].
**EMT**				
	Nicotine induced EMT in different epithelial cells (Reviewed in [12,83] and [99])	SARS-CoV-2 infection shifts cells to a more mesenchymal phenotype, which is confirmed by downregulation of EPCAM expression following viral infection [81].		
E-Cadherin	Decrease of E-cadherin (this work and [99,100])			
Fibronectin	Increase FN (this work and [101,102])	Bioinformatics analysis shows that FN interacts with ACE2, the mRNA and protein of this molecule are more expressed in lung epithelial cells after SARS-CoV-2 infection [103].		
Vimentin	Increase Vimentin (this work and [104])		Important role for Vimentin in SARS-CoV virus entry through interaction with its S protein [105] in the SARS-CoV-permissive cell line Vero E6. In these cells the expression of extracellular Vimentin is upregulated after virus interaction and enhances its delivery to ACE2.	
Cell Migration	Increase cell migration (this work and [12]).			
Anchorage independent growth	Increase slowly (this work and [106])			
**Neo-Angiogenesis**				
	Increase (this work and [83,106])	SARS-CoV-2 infection show the presence of IA as well as of conventional SA [107].		Viruses, such as CMV or HCV may regulate angiogenesis directly or indirectly, may activate vessels through endothelial cell tropism and/or producing chemokines and/or growth factors (i.e., VEGF) creating a pro-angiogenic microenvironment [108].
VEGF	Increase VEGF (this work and [83,109])	Bioinformatics analysis shows that VEGF interacts with ACE2, the mRNA and protein of this molecule are more expressed in lung epithelial cells after SARS-CoV-2 infection [103].		
**Effects Induced by Nicotine in SARS-CoV-2 Infection**				
		Increase SARS-CoV-2 replication in A549 cells [47].		
		Increase the transcription of SARS-CoV-2 viral proteins in A549 cells [47].		
		Increase SARS-CoV-2 cytopathic effect in A549 cells [47].		

## Data Availability

The data presented in this study are available on request from the corresponding author. The western blotting images presented in this study are available in supplementary material.

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
