# Peer review of "Nicotine Changes Airway Epithelial Phenotype and May Increase the SARS-COV-2 Infection Severity"

_molecules, 2020, doi:10.3390/molecules26010101_

Round 1
Reviewer 1 Report
The study of Lupacchini et al is devoted to the investigation of the nicotine effect on respiratory cell growth taking into account the possible involvement of nicotinic acetylcholine receptors in SARS-COV-2 cell infection. The article is in general, well-written, the experimental design is appropriate and the main article message is supported by experimental results. Also, an impressive analysis of nicotine and SARS-COVs influence on airway epithelium is presented. However, some issues should be addressed before article acceptance. Major:
- The article title is very “general” and should be clarified. For example “Nicotine changes airway epithelial phenotype and may increase the SARS-COV2 infection severity”.
- Please, shorten the introduction (lines 40-70). Maybe, some information can be included in the manuscript as Appendix A.
- You stated that Nicotine has a greater affinity for a4b2 receptors than for a7 nAChRs, while your experiments here and in ref32 are based on the idea that a7-nAChR mediates main nicotine effects. Could you justify that? Is it possible that in airway epithelium a4b2 nAChR also can be mediate some nicotine-induced events and thus be also implicated in SARS-COV-2 infection? To check that silencing of a4b2 should be performed or a4b2 antagonist (such as DHbE) should be co-applied with nicotine.
- Please divide all figures into panels (a, b, etc.) and show statistical analysis on every panel (not only p-values but asterisks too). Note, that the 2-sided t-test is not exactly correct to analyze more than two data massives (for figs 1,2,4,5,6,8,9,10,11). One-way ANOVA with multiple-comparison is recommended.
- For Fig 2, 4,5,7, and 10 the calibration should be shown as a linear regression line. Please, do not forget to check regression equation linearity (this can be done in Prism via “test departure from linearity…”) and show your linear regression equation on the figure.
- Recently, several works describing the relationships between human three-finger regulatory proteins expressed in lung tissues with a7-nAChR upregulation under nicotine treatment and upregulation of different intracellular pathways inclidung MAP/ERK, p53, and p38 were published (Bychkov et al, 2019, Shulepko et al, 2020). Please, involve the discussion of possible correlation between expression level of different endogenous three-finger regulators with a7-nAChR and nicotine roles in SARS-COV-2 cell infection.
Minor:
- Line 134-5 “does not EXPRESS a7-nAChR or ACE2 proteins”. In ref32 you have shown that nicotine does not induce ACE2 UPREGULATION in cells with silenced a7-nAChR but a7-nAChR silencing does not change ACE2 expression, please clarify your point.
- Please, show your data for the total p38 expression evaluation
- Please, decipher main abbreviations, such as HBEpC, nAChR, Ach, ACE2 in the main article text.
- Provide dilutions for all antibodies used in the study, including secondary ones.
- Line 201, 212, 410. Did you mean “row” or “raw”?
Altogether, the level of the article is high enough to be accepted for publication after minor revision.
Author Response
Replay to reviewer n. 1
Dear Reviewer,
many thaks for your kind comments and for suggestions. In the revised manuscript in red you have all the modifications.
- We changed the title according to your suggestion in "Nicotine changes airway epithelial phenotype and may increase the SARS-COV2 infection severity"
- We shorten the Introduction and we introduced a new Table 1, reporting all the properties of Nicotine
- We explained that alpha4 is essentially related to addiction. A high affinity implies that the receptor is "freezed" after Nicotine treatment ad no more able to work. In lung cells the most important player is alpha7. In the supplementary file we reported blots performed in cells with alpha4 silenced. In these cells after nicotine treatment ACE2 does not increase.
- All the figures are divided in panels and both p values and asterisks are reported.One-way ANOVA with multiple-comparison is reported in the revised figures.
- Calibration as a linear regression line is performed and linear regression equation was put on each pertinent figure.
- The suggested works describing the relationships between human three-finger regulatory proteins expressed in lung tissues with a7-nAChR upregulation under nicotine treatment and upregulation of different intracellular pathways inclidung MAP/ERK, p53, and p38 were added and discussed on the ligth of SARS-CoV-2 infection
Minor:
7. We clarified that in a7-si-mRNA after nicotine treatment ACE2 did not increase
8. Data for the total p38 expression evaluation was added in the new Figure 8
9. A list of abbreviations is at the end of the manuscript
10. The dilutions for all antibodies used in the study, including secondary ones are reported in the section Materials and Methods.
11. Sorry for the error we mean raw and it was corrected.
Reviewer 2 Report
Lupacchini and colleagues report an extensive study on the effect of nicotine in SARS-CoV-2 infection. Authors conclude that is likely that nicotine is involved in SARS-CoV.2 infection. Although conclusion seems to be proved by the results, I would suggest some changes that will allow a better presentation of data.
First, manuscript is not clearly written. There are very long paragraphs in the introduction, spelling errors and numerous unspecified abbreviations throughout the manuscript. It needs a revision.
Second, the reasoning of why the successive experiments are being carried out is almost non-existent, there is no common thread.
Third, figures are too small to clearly see them. Ordenates axes should specify what measurements are. Statistics explanation should accompany each figure.
Author Response
Dear reviewer n. 2
Many thanks for your observations and suggestions.
The introduction was shorted and a new Table 1 with all the properties of Nicotine was introduced, spelling errors were corrected, abbreviations are listed in the abbreviations list at the end of the manuscript.
Second, the reasoning of why the successive experiments were carried out is reported in the new introduction at the end.
Figures are enlarged. Ordenates axes specified what measurements are. Statistics explanation is in Materials and Methods. And all figures show the statistical evaluation. p>0.05 is not significant
All the corrections are in red, also corrections required by the reviewer n. 1